

# GC Insights: Communicating Climate Change – Immersive Sonification for the Piano

By Charles Jahren Conrad
charlesconrad1000@gmail.com
Blindern Videregående Skole IB, 0855, Oslo Norway



# *Abstract*


In order to convey climate change to a wider audience, I converted various $CO_2$ records (parts
per million) into music for the piano (scale notes) through the method of sonification. This is
a data driven piece with five movements and includes musical elements such as tone, chords
and key signatures, along with the data driven notes, providing a sonic experience of climate
change and the acceleration of emissions. Because this composition can be played on the piano,
it provides a level of immersion beyond a visual or auditory understanding, conveying the
urgency of climate change to a broader audience in a new way.

# Introduction


The goal of this project is to raise awareness of climate change and the effect that climate
change has upon the environment and the world. The most significant indicator of climate
change in our modern world is carbon dioxide, $CO_2$ (United States Environmental Protection
Agency, 2020). It is therefore vital to spread this data and encourage new engagement
through new medias, such as sonification. I implemented the methodology of sonification
(the use of non-speech sound to convey data and information) to mathematically transpose
$CO_2$ records, recorded at Mauna Loa Observatory, into musical notes that are playable on the
Piano (Earth System Research Laboratories, 2020). This medium of music increases
accessibility, audience, and memorability, providing the immersive experience of climate
change that the data deserves. I sought out to create a new type of sonification, that combines
statistics and creativity to provide audiences with an original, immersive, and enjoyable
musical piece that is still statistically accurate, resulting in the *Statistical Composition.*



# Sonification Use and Effect

In this project, I use auditory display, a form of sonification where the purpose is for the audience to listen to a piece of sonification with high index (high sonification accuracy to original data). I mapped data into sound through set parameters and boundaries. This is most notable in one playing hand, often the right hand, of the piano piece throughout the five movements of the Statistical Composition. Something unique about this Statistical Composition is the important creative elements that incorporate sound art and composition. This is sonification that is inspired by the mathematical conversions, however this component has a relatively low index. In the piece, the other hand, usually the left hand, will play notes that are creatively composed to balance the piece, add nuance, emphasis, and emotion to certain parts of the piece, and to create further immersion and audience. This combination provides a full and immersive experience while retaining a level of accuracy to the data set through parameter mapping.

This project and the method of sonification increase scientific accessibility to those that are less able. The fact that it is an auditory medium, provides not only a new and novel experience as mentioned previously, but it is helpful for those that are visually impaired, thus engaging a broader audience. Sound is considerably more memorable compared to data and complicated information (Mair, 2022). My sonification project uses six elements of sound, linear time, varying length of certain notes, frequency, amplitude, rhythm and even type of instrument or sound, thus conveying more information than a multi-dimensional graph.



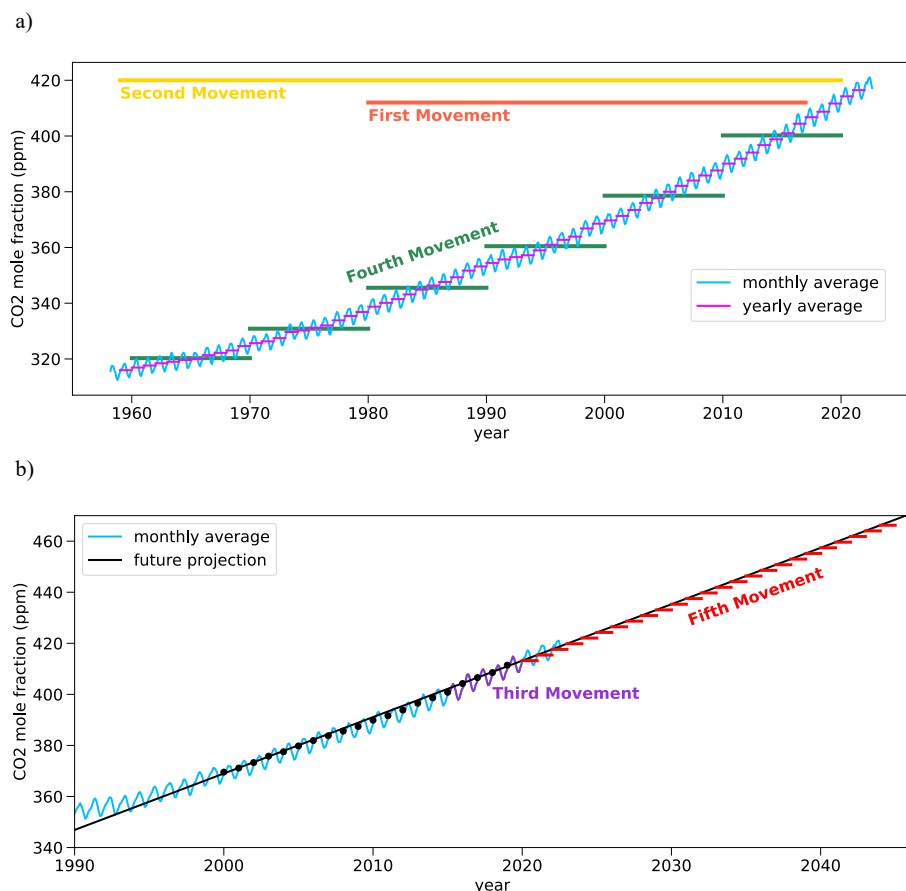

54

**Figure 1**: Statistical Composition methodology, found at:

*https://www.dropbox.com/s/ixvineu1mmdi2om/Statistical%20CO2%20Composition.mp3?dl=0*

a) $CO_2$ levels (ppm) at the Mauna Loa Observatory with first, second and fourth movement

labels. b) $CO_2$ levels (ppm) with third and fifth movement labels. Incorporated is the monthly

averages from 1990 alongside future projections to 2044.


# Methodology: Numbers to Notes



The five movements used to separate periods of the data followed a basic common procedure
of range differentiation with common musical backbone. The entire composition is set to the
time signature of $\frac{4}{4}$, or a quarter note having the value of a quarter of a measure, and the key
of middle C. The technique of range differentiation based itself on finding the difference
between the highest $CO_2$ value, which was always the more recent value chronologically, and
dividing this by the number of piano keys that were desired to be included. This assigned a
piano key/note a respective $CO_2$ value, however this would differ between movements (see
figure 1, a and b). Table S1 (supplementary material 1) summarizes the fundamental
differences in the base parameters of each movement.

**Movement one: 40 years of yearly increase**
In the first movement, yearly values were mapped to the closest notes, played by the right
hand. To accompany the right hand, the average $CO_2$ ppm value for every four years was
calculated. The preset note values from the previous range differentiation calculations were
used to convert these new four year ppm values into notes, thus using the same conversion
method for both the right and left hand which would play together. These notes were set an
octave down to create space for both hands to play simultaneously. The left hand played
chords of whole notes, with the root note being the four-year average (see figure 1 a).

**Movement two: A complete 60 years of climate change**
The second movement was based off yearly averages from 1959-2019, and the value had to
exceed the closest note value, promoting positive change. Instead of having the same notes
repeat themselves, repeating notes were joined together and created a longer sustaining note,
thus creating a varied and complicated rhythm with the intention of making the piece more
varied and rhythmic. For the left-hand, major chords were composed that had the same root



note as the right hand, still based off of the data set. These notes were also set down an octave
however they were parallel in rhythm and note.

**Movement three: Monthly fluctuations in recent five years**

The third movement relied on the change of $CO_2$ value, so if the change was less than 0.5
there would be no change. However, if the change was greater than this and equal to or less
than one, there would be an increase of 1 note. If the change was negative, this was mirrored,
and the possible change would be downwards. This note conversion created the left hand, and
the right-hand plays minor (dark and dissident) chords that have the root note of the first
quarter note in each measure.

**Movement four: Decade averages**

The fourth movement had a set step for each note of 5 ppm, which defined the root notes of
chords that were built creatively around this direct conversion. The left hand plays these
minor chords that have the root note of this value creating a similar effect as in movement
three. An octave above, the right hand arpeggiates these minor chords that the left hand plays
creating an interesting and quick rhythm, doubling in speed in the last two measures before
the final A minor chord concluding in an additional A note. The arpeggiating chords create
urgency in the music.

**Movement five: Yearly values of the 21ˢᵗ century**

The fifth movement was a projected $CO_2$ rise, and therefore differed from the rest of the
movements. Using the previous data, a curve was plotted by averaging out the data set and
creating the best fit line, $y_{(CO2\ ppm)} = 2.21x_{(years)} - 4052$, used to estimate future $CO_2$ levels.
Notes were assigned to values with an even amount of rise per year. These notes were the



root notes upon which the chords are built. Major chords for natural root notes and minor
chords for sharp or flat root notes was the utilized pattern. The left hand would mirror the
right and play exactly these two octaves down. With the sustain pedal added, the movement
created a futuristic effect which truly differentiates it from the other movements.

## Ethical Statement


No ethical statement was needed as no interpersonal contact occurred and all data was
retrieved from the public domain.

## Conclusion and Implications


The Statistical Composition incorporates $CO_2$ ppm values recorded from Mauna Loa
Observatory from 1959-2019, both yearly and monthly averages. It also incorporates
projected CO2 values until 2044. This creates a uniquely playable piano song that accurately
represents $CO_2$ levels rising, and thus the rise of climate change and global warming. By
composing an accompaniment, a full, broad, and dynamic song is created that accurately
portrays climate data.
The implications of this product can be viewed in a broad spectrum. I converted data and
statistics, only available in English, into a musical piece that is translated into the language of
music that anyone in the world can understand, regardless of what language they speak. By
combining creative and logical thinking, a new experience was both calculated and created,
providing a unique musical and scientific experience. This piece differs from conventional
sonification in the sense that it is physically playable and directly experienceable by a wide
musical audience.




Link to the statistical composition via *Dropbox*:
https://www.dropbox.com/s/ixvineu1mmdi2om/Statistical%20CO2%20Composition.mp3?dl=0

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
