# Peer review of "GC Insights: Communicating Climate Change – Immersive Sonification for the Piano"

_EGUsphere, 2022_

## Author Comment (AC1)

**Response to Reviews of *EGU* manuscript EGUSPHERE-2022-1356**
**February 11th, 2023**

**Response to Reviews of *EGU* manuscript EGUSPHERE-2022-1356**
**February 11th, 2023**
**Title: GC Insights: Communicating Climate Change – Immersive Sonification for the Piano**
**Author: Charles Jahren Conrad**
* * *
RC1, Referee #1 comments and my responses (in boldface and blue text):

General comments:

Thanks for your contribution. A common pitfall of this kind of musification is that the music winds up sounding similar. Basically, if CO2 or temperature is rising constantly and you link it to pitch, all the resulting music will sound roughly similar. However, by only linking one hand to the data while allowing the other to perform original music, you're doing something unique and you've managed to side-step that pitfall - so congratulations!

I hadn't encountered the GC-Insights type or submission before, so I realise that some of my comments may not be addressed within the format this kind of article. For instance, the manuscript doesn't strictly follow a scientific article template, ie it has no results or discussion sections. I'll defer to the editor to confirm whether they are required for GC Insight publications.

I'd like to see some comments on previous work on sonification of climate change data in your introduction. Typically, references don't contribute to the total word count, so you should be able to add as many as you'd like. Here are some starting points:

- Borromeo, L., Round, K., and Perera, J.: Climate Symphony, available at: https://www.disobedientfilms.com/climate-symphony 2016.

- Crawford, D.: Planetary Bands, Warming World string quartet, Video published by Ensia magazine, available at: https://vimeo.com/127083533, 2013

- the Climate Music Project (https://climatemusic.org/ )

- de Mora, L., Sellar, A. A., Yool, A., Palmieri, J., Smith, R. S., Kuhlbrodt, T., Parker, R. J., Walton, J., Blackford, J. C., and Jones, C. G.: Earth system music: music generated from the United Kingdom Earth System Model (UKESM1), Geosci. Commun., 3, 263–278, https://doi.org/10.5194/gc-3-263-2020, 2020.

It would be good for you to use these to highlight how your work is novel and different from previous approaches.

**Thank you for the feedback and manuscript improvements, and for pointing me towards this relevant literature. I agree with your suggestion as to the value of commenting on previous climate sonification work. These references are suitable for verifying the impact of such climate sonification that I outline, and also to emphasize the uniquely playable trait of my sonification piece. Such references will highlight the individual immersion into the CO2 data that is possible through my piece. I will incorporate these into the manuscript.**

The audio file:

- There are no clear breaks between the five movements. Perhaps a fermata and a bar of rest between them might help separate each movement?

**Distinguishing the five movements from one another is vital within the paper, and should be easily distinguishable in the piece itself. In producing the digital audio file through *Logic Pro X*, I will add a measure of rest in between each movement and place a fermata on each movement's final note. This will also add a level of personal involvement for potential interested musicians, as they can decide how long this fermata lasts, effectively breaking up the movements. This will be described in the methodology additionally.**

- The syncopation of the first movement makes it harder for me to perceive time passing. I think perhaps you could decouple the rhythm of the left and right hands such that the left hand is closely linked to CO2, but the right hand anchors the time signature. (This is an artistic choice so I leave it up to you whether this improves or deteriorates the piece.)

**I understand this comment and its purpose in increasing timeline clarity for an audience. In my artistic evaluation of the piece, I believe that this would increase the difficulty of playing the piece and limit the clarity of the CO2 rise. Having the right and left hand play different rhythms throughout the movement increases the compounded nature of the lower tones and pitches prominent in this movement, as the right and left**

hands would play at different times and often over each other. This would especially jumble the movement, and take away the focus of the CO2 data which especially in the introductory movement of the piece, is intended to be highlighted. Therefore, in my artistic evaluation, in order to keep the same level of CO2 rise clarity and to limit the degree of difficulty of the piece, I suggest that this change not be implemented.

- I'm not a huge fan of the sound of this instrument - it sounds very dry and digital. Perhaps a different virtual instrument might produce a better sound - or alternatively you may be able to use some reverb and Eq? If a huge budget were available, then you may be able to find a local recording studio with some expensive microphones and a grand piano you could use to record your performance. Or maybe a pianist on a service like Fiverr could perform and record it for you?

Unfortunately no budget is available for this project, so recording studio sessions and a pianist from a service is unrealistic. That being said, it is important to make the piece as auditorily pleasing as possible. I personally consider the Steinway grand piano the best standard virtual piano within *Logic Pro X*, however I fully concur with your assessment of the dry and digital sound. I have added reverb and Eq to this piano as suggested in order to address this area of improvement. Using *Chroma verb stereo*, I have added reverb to the middle and most dominant frequency ranges of the piece. I have set the wet factor to 50%, decreased dryness down to 85%, added a decay time of 1.1s, and set size and density to 60%. Additionally, I have Eq the sound, limiting the lowest and highest frequency ranges of the piece. I chose to limit these extremes as the audio sounds most digital when low notes are being played together and when high notes are played in unison. The sound seems digital and distorted at these extremes and with these improvements, the piece is immediately more enjoyable. I have included a dropbox link (I realize that this is only a preliminary place to keep this piece online) of the Statistical Composition in its edited version to show these changes:

https://www.dropbox.com/s/qsnkg6hy763rapf/Statistical%20Composition%20%28Edited%29.mp3?dl=0

I don't think that dropbox is the best place to keep a permanent record of this piece. The first place would be to append it to this article as a supplementary file. A scientific data repository might also be appropriate, something like zenodo or BODC, plus this would provide a DOI. As a backup, youtube or soundcloud or might also work for hosting, however it's not guaranteed that any of these companies will exist in ten years (including Dropbox).

**I believe the scientific depository of *Zenodo* would be a suitable site to provide long-term access to the digital audio file and also the sheet music of the statistical composition. I have created an account and determined that this process would be straightforward and manageable when the changes to the piece are finalized, and I thank the referee for this valuable comment.**

I'd like a section on how the recording was created as well.  Did you program the MIDI and pass it to a virtual instrument or did you record a live performance? What instrument, microphone and interface (if any) were used? What VST have you used to generate the audio? Did you use a DAW, if you which one? Were any post-processed effects added? reverb, compression, delay etc. Was any mastering applied?

**This comment is highly valued as including such a section on the digital aspect of the methodology will enhance the detail and exactness of how this piece was truly constructed. The DAW that I used was Logic Pro X, however when composing the notes around the data converted notes, I used the AKAI Pro MPK mini play keyboard. These notes were recorded digitally and then their rhythm and note value were confirmed and corrected within Logic Pro X. This MIDI keyboard connected directly to the computer running Logic Pro X, and no interface was necessary. Logic Pro X has the incorporated instrumental plugins and VSTs, and I used the Steinway Grand Piano in this manner to produce the digital sound. As for the submitted manuscript, no post processed effects or mastering was used. However, I have incorporated reverb, Eq and delay into the song as discussed above. What I outline here will be incorporated into the methodology section of my manuscript, improving the clarity and preciseness of the GC Insight greatly. Some of these comments may perhaps go into the supplemental considering space limitations of the GC Insight format.**

The main criticism that I have of this draft is that the author does make quite a few unsupported statements in the abstract, introduction and conclusions. I've made some suggestions here, but I'd recommend a careful re-reading, to ensure that what is written is accurate, and not hyperbolic.

**This is a completely valid statement. I am still learning to improve my academic communication, and instances of hyperbole and inefficiency are too prominent within my writing. I welcome all comments on this front as this will greatly improve the manuscript. Upon re-reading the manuscript, I too notice instances of lack of clarity and exaggeration, and I will continue to check for such disparities.**

A second criticism is that there's only one image permitted in Insight articles, so you really need the figure to shine. You could have one pane about the sonification method, one pane about the recording method, one about the data derivation. At the moment, this figure is not very clear and it would really be worth putting in the effort to make it great.
On the whole, I'm happy with this as an Insight article, and I enjoyed the music.

**Thank you for the detailed, thorough, and helpful comments and evaluation. The figure is definitely an important attribute to this manuscript. I have outlined how I will improve this farther down in this rebuttal letter.**

Specific Comments:
Abstract:
L11: remove (parts per million)
**Completed.**
L12: remove  (scale notes)
**Completed.**
L12-:L15:  This entire sentence should be replaced with a brief but explicit characterisation of your method. Something like "CO2 measurements from Mauna Loa were linked to musical pitch to drive the sonification, but additional musical parts were creatively composed to balance the piece, add nuance, emphasis, and emotion to the piece."  (This is the part of your work that really stands out to me: it's not 100% data driven, and the musical freedom that you allowed yourself makes it stand out. It's worth emphasising this in the abstract! )

**I agree with this improvement as it specifies the "sonic experience", introduces the origin of the data from Mauna Loa, and gives good overview of the insight. I will complete this improvement.**

L15: Because -> As

**Completed.**

L16: I'm not sure this is true: "it provides a level of immersion beyond a visual or auditory understanding". However, I do agree that it certainly adds a sense of urgency and gloom to the data.

**Thank you and I have altered the final sentence of the abstract as such: "it encourages engagement while adding a sense of urgency and despondency through conveying climate change to a broader audience in a new way."**

Introduction:

L20: If the goal of the project was to raise awareness of climate change, how do you do that? Have you tracked the number of listeners or shown where they came from? Were they already aware of climate change? To me, it looks like the goal was to generate and share a piece of music based on climate data.

**Such impact tracking has not been carried out for this project and therefore this comment is very much appreciated. This sentence has been rephrased as follows: "The goal of this project is to create and share a piece of music based on climate data."**

L21: CO2 isn't an indicator of climate change - it's one of the main causes.

**I have changed indicator to causes.**

L23: Climate change is pretty well established at this stage.  right?

**I agree with this. The intent was to justify the use of new methods however this does not need to be outlined clumsily. I have made this more direct: "New engagement through new medias, such as sonification, is useful for conveying this dangerous trend."**

L25:  remove "mathematically"

**Completed.**

L26: remove " that are playable on the Piano"

**Completed.**

L27-L29: This is unsupported.

**I have included the reference used later in the manuscript of a lecture from Karen Mair in 2022 on Sonification for Geoscience (Mair, 2022) in order to justify this.**

(Mair, K. Sonification for Geoscience Turning data into sound: https://www.youtube.com/watch?v=gh02Tb94oHs, last access 21 June 2022.)

This can be further evaluated using the following source that you have suggested:

• de Mora, L., Sellar, A. A., Yool, A., Palmieri, J., Smith, R. S., Kuhlbrodt, T., Parker, R. J., Walton, J., Blackford, J. C., and Jones, C. G.: Earth system music: music generated from the United Kingdom Earth System Model (UKESM1), Geosci. Commun., 3, 263–278, https://doi.org/10.5194/gc-3-263-2020, 2020.

Referencing this source would validate Sonifaction's impact towards accessibility and engagement to non-experts.

L29: remove "out"

Completed.

L29: Is this really a new type of sonification? There is definitely a precedent of other people combining data and musical choices.

Defining this as "composing a unique sonification" improves the clarity of this. This shows that I did not create a new style, but composed a unique piece instead. In response to a comment from RC2, I am citing two sources of sonification that uses a similar style of physically playable sonification.

L30: I don't understand how statistics got involved here or what is meant by statistically accurate? These are specific terms that don't fit this context. I recommend changing this to: "combines climate data and creativity", and "musical piece that is data driven"

These changes have been applied and I thank the referee for the improvements to my communication.

Sonification Use and Effect

L34: " auditory display:" (replace , with :)

Completed.

L35: remove " high index (" and following ")"

Completed. I have implemented the term indexicality and sourced the sonification handbook: Indexicality is a measure of arbitrariness of data mappings, and high indexicality depicts a large degree of conversion accuracy (Hermann, Hunt, & Neuhoff, 2011). (Hermann, T., Hunt, A., & Neuhoff, J. G. (2011, November). *The*

*Sonification Handbook.* Retrieved February 11th 2023, from Sonification.de: https://sonification.de/handbook/)

L47: remove "to those that are less able"

**Completed.**

L48-50: unsupported statement.

**I have decided to remove this unsupported statement as it does not greatly enhance the argumentation that is later supported through references, and there is little evidence of climate sonification greatly improving accessibility for those who are visually impaired.**

L52: What do you mean type of instrument? I only hear a piano.

**This has been removed. Additionally, I agree there is little distinction so varying length has been removed, along with instrument as only the piano is used. It has been changed to: "My sonification project uses four elements of sound, linear time, frequency, amplitude, and rhythm, in creating the *Statistical Composition*."**

L52: Might be worth reading and references Flowers 2005 here. The key thing to note is that it's actually quite hard to get a lot of information out of sound, especially as with a single instrument you can't modify the tone, and it's challenging to perceive small fluctuations in amplitude. (Flowers, J. H.: Thirteen years of reflection on auditory graphing: promises, pitfalls and potential new directions, Proceedings of ICAD 05-Eleventh Meeting of the International Conference on Auditory Display, Limerick, Ireland, 6–9 July, 406–409, 2005, http://sonify.psych.gatech.edu/ags2005/pdf/AGS05_Flowers.pdf )

**I agree with this helpful comment that referencing this adequately to address both positives and negatives of the use of sonification. This will be implemented both here in the introduction and later in suggested improvements.**

Figure 1: This figure is not very clear to me. Did you use monthly or annual data? Why are movements 1 and 2 shown as straight lines, but movement four is segmented? Third movement uses monthly data? I think you would be better served by having five panes, one for each movement, and showing the Mauna Loa monthly data in black, and the values that you used to drive the modification as separate coloured lines.

**This is a very helpful comment. I have changed the figure as follows, and included the changed figure in this response below. I have separated the panes and made each movement distinct within its own pane. I also included a screenshot of the Logic Pro X**

**methodology as outlined in your comment for the sonification method. To incorporate the data derivation into a figure, I have added notes saying that a change in CO2 converts to note value, and that a change in time converts into the song's rhythm. These notes are included along the axes of the Logic Pro X still (part b of the figure).**

[Figure]

**Figure 1:** *Statistical Composition* **movement data and audio methodology. (a) CO₂ levels (ppm) at the Mauna Loa Observatory, showing (a.0) monthly data and deseasonalized trends, and (a.1-a.5) the incorporation of this data in the respective movements of the piece. (b) Audio methodology showing a still from the Logic Pro X piano roll (left) and the resulting score (right) with data sourcing descriptors on the axes.**

L55: this isn't really the methodology, it shows which sections of the data were used by the sonification.

**I have changed this, stated above.**

L56: you don't need the link to the dropbox file here.

**Completed.**

Methodology: Numbers to Notes:

L62: remove "basic":

**Completed.**

L63: I've never heard of a " common musical backbone". Can you elaborate on what this means?

**This means a time signature of $\frac{4}{4}$, a quarter note having the value of a quarter of a measure, and the key of middle C. This is defined in the next sentence however this is not clear and will be made so. I have also addressed this in response to referee 2, RC2.**

L72: We typically use "annual" instead of "yearly", but as this is the title of the movement, it's an artistic decision.

**I agree with this use of common terminology and have changed the title of the movement to "annual". This has also been completed throughout the rest of the manuscript.**

L72: For this and the other movements, please indicate at what timestamp they begin in the recorded piece.

**This is understood and will be changed.**

L82: " and the value had to exceed the closest note value, promoting positive change": What does this mean - can you make it clearer, please?

**This has been changed to "All values were rounded down to their respective notes, which slightly reduces the CO2 emissions that are conveyed in the piece."**

L98: Decade -> Decadal

**Completed.**

L109: Is there any reason why you fitted to recent data rather than using established CO2 projections (SSP5-8.5 or even RCP8.5 would both be appropriate. ) Ultimately, I suspect the difference is small, but you may reach a wider audience using these well-established projections.

The reason this method was used was to show how CO2 emissions would increase if the rate of change stayed constant, something that history has proven to be untrue as this rate itself increase. My use of a fitted future projection is effectively showing an optimistic projection that still creates urgency and despondence in the rising CO2. This can also be discussed in a challenges and future work section. I agree that it would be worth comparing to these established CO2 projections.

I am including my response to RC2 surrounding this same area of improvement:

"My prediction is closest to scenario SSP1-2.6, an optimistic projection of low future emissions. This is visualized in the figure below, however due to copyright this figure will not be included into the manuscript. I will however source this projection and compare 2044 values with my projected CO2 levels and the scenario's. I will source this database here as it is more recently updated:

https://www.ipcc.ch/report/ar6/wg1/about/how-to-cite-this-report/. "

[Figure]

L124:  uniquely playable ->  unique and playable

 Completed.

L124:  piano song -> piece for piano

 Completed.

L126: song -> piece

 Completed.

Ethical statement

The ethical statement should be after the conclusions.

**Completed.**

Conclusions:

L129: "only available in English": I don't think that Mauna Loa data is in English! It's just Arabic numbers!

**I have deleted this statement, and yes I agree with this improvement. The cite itself is mostly English, however I realize that it can be translated and that my manuscript is itself in English as well. Interesting to think about how both numbers and music are universal to a certain degree!**

L130: This is a bit of a bold statement: "anyone in the world can understand, regardless of what language they speak". It's not clear to me that it's true. I'm not sure that this piece would make sense if you just heard the music. In order for it to mak sense, it needs to be explained in context that it is derived from climate data.

This is understood and agreed with.

L128-130: To be honest, I think you can safely remove the first two sentences of this paragraph.

**I agree with these comments and recognize the hyperbole in these sentences. These sentences have been fully removed to avoid confusion and hyperbole.**

L132: "providing a unique musical and scientific experience." While this is indeed a unique experience, it's not what I would focus on here in the conclusions.

**This has been removed.**

I'd like to see some suggestions on potential improvements. Ie, alternative datasets, audience survey, etc. See for instance de Mora et al, mentioned above.

**This is a very sound comment on a truly lacking part of my manuscript. Potential improvements would be (as outlined) a survey to estimate impact and the function of my piece, the use of alternative data sets, and incorporating a visual animation that depicts the data's rise in value in coalition with the playing of the piece. This will be included prior to the conclusion. I will add these suggestions for ways to move forward with this particular type of study. For example, does learning and playing the piece actually give an improved understanding of climate change?**

Supplement:

Table: Please add a caption or a description of the table.

**A sufficient caption will be provided.**

Sheet music:

- Please add the tempo

  **This will easily be implemented.**

- Please add the instrument (piano)

  **This will be specified.**

- You may want to add notation of when to hold and release the pedal.

  **This shall be included into the score.**

- Please indicate where each of the five movements begins and ends. I'd recommend a double bar line at the end of each movement. as well as the title of the moment (ie Movement one: 40 years of yearly increase).

  **I concur with this improvement and I will correct this.**

- This would also be a good opportunity to clarify where data came from directly in the music. Ie notes on the pdf statring "right hand plays annual mean CO2 from 1960-2015' or similar.

  **I agree that this provides an opportunity for sufficient structure in the movements and piece. This will be completed.**

**These improvements will improve the supplement and my communication. The separation of the movements is vital and is therefore quite necessary.**

**I thank the referee for each of these helpful comments. I am motivated to address each area for improvement within my manuscript and greatly appreciate the detail in which the referee evaluated my work.**

---

## Author Comment (AC2)

**Response to Reviews of *EGU* manuscript EGUSPHERE-2022-1356**
**February 11th, 2023**
**Title: GC Insights: Communicating Climate Change – Immersive Sonification for the Piano**
**Author: Charles Jahren Conrad**

RC2, Referee #2 comments and rebuttal (in boldface and blue text):

General comments

The author proposes a musically-oriented sonification of CO2 concentration data. The musical piece is partly data-driven through the method of "parameter mapping", but some aspects of it (chords, dynamics) are based on a musical decision. This can be thought of as some form of data-based composition or musical arrangement. The intent is to facilitate the communication of such important data to the general public through an engaging musical experience.

The musical piece itself is interesting and quite intriguing to my ears, due to the very "chromatic" approach taken here and the way major/minor chords are used and combined. Moving beyond a purely data-based sonification and using the freedom brought by musical composition is an interesting aspect of this work in my opinion: it is both powerful (because the composer can induce intent and emotion) and challenging (because the link to the data might become weaker).

In terms of objective, I agree with the author that musically-oriented sonification has a great potential to communicate data and concepts to the general audience. One specific challenge I can foresee for this piece is that the 5-movement structure makes the music "play the data" several times, but for different time periods and data resolution (yearly/monthly): this is difficult to follow based on the audio only. The figure provided in the paper is helpful in this respect, but I believe some form of data animation would be much more efficient - maybe a future improvement worth discussing?

The main criticism I have relates to the way the paper is written: it could significantly be improved in my opinion. If this paper was to be considered as a regular scientific paper, the

main issues would be the following:

1. A review of previous works is missing in the introduction, which makes it difficult to understand how the author's piece compares to existing works.

2. Explanations on how the musical piece was created and arranged are not very clear in my opinion.

3. There is no discussion on the challenges faced to create the piece, possible improvements, future works etc.

I understand that is not a regular scientific paper, but still, I think improvements in the three areas above would be useful to the reader.

**Thank you for the extensive, constructive and helpful review. I intend to fully address all of these comments and improvements, and to implement the overall majority of them. For the three main issues that the reviewer mentions above, I plan to make the following changes:**

**1. First of all, I agree with a lack of recognizing and referring to previous work and literature around the topic of climate sonification. This is an important part of the scientific process that will further validate the uniqueness of my piece and the effectiveness of sonification itself. I will add more references to previous work, as I described in my response below and to RC1.**

**2. I agree that the methodology within the production of the audio file was not fully developed and explained. To improve this I will delve deeper into my use of Logic Pro X (see details about this in my response to RC1) and incorporate a still from that methodology into the figure (see part (b) of the new figure that I propose in my response to RC1).**

**3. I agree that my paper would greatly improve with a discussion as to the challenges, possible improvements and future extensions and unanswered questions. This is something I plan to expand upon. I think that developing a data animation to go along with the audio file is beyond the scope of this work, but it would be an important**

**extension of this study and I will discuss that, along with other possible future improvements, in the conclusions (also see my response to RC1).**

Specific comments

Throughout the paper: Climate change and CO2 concentrations are two distinct things - the latter is one the main driver of the former. This distinction could be made more explicitly in the title (e.g. "Communicating the Causes of Climate Change" or something along those lines?) and throughout the paper. As an illustration on line 22, I don't think CO2 concentrations can be termed an "indicator of climate change": global temperature (for instance) would be. In fact, an interesting sonification experiment would be to play both the cause (CO2 emission) and the consequence (increasing temperature) together: worth a word in the discussion?

**I have changed the word "indicator" to "cause" in the introduction. This distinction is important and thank you for pointing this out. I have corrected this distinction throughout the rest of the manuscript as well. I agree that changing the title to "Communicating the Causes of Climate Change" would increase the accuracy of the title. I have changed this as such, however to fully change the title, a longer process may perhaps ensue in correspondence with the editor. Playing both temperature and CO2 change would definitely be an interesting insight and possible extension of my project, and I will include this in the final section of suggested improvements and unanswered questions.**

Lines 29-31: could the author explain a bit more the choice of using the adjective "statistical" ? Is it only because the process is based on data, or is there additional intended meaning? Note that I'm not disputing this choice: sonification is a way of presenting data, and that's indeed part of Statistics, but it might be worth stating this explicitly because I reckon some readers might have a narrower interpretation of this term.

**I have changed this to climate data instead as this is more understandable and applicable to this specific line. I will however incorporate a distinction within the definition of sonification as to its relevancy towards the CO2 data itself. I use the word statistical in the name of the piece, Statistical Composition, to convey connotations of an artistic and mathematical song. This is what my work is in essence. Distinguishing this use of statistical is definitely a valued comment, and I will implement a definition that**

**the piece is a way of presenting these statistics and data. Thank you for this helpful comment.**

Lines 20-31: this introduction does not do any literature review. To start with, I think the sonification handbook (https://sonification.de/handbook/) is worth citing for all technical aspects behind sonification, and also possibly for discussing topics related to auditory perception. In addition, the author could mention other sonification works and discuss similarities and differences with the presented piece. The list below is not exhaustive but provides a few examples for which I found similarities with the author's piece (in terms of either underlying data, creation of a physically playable piece or inclusion of subjective composition elements).

* $CO_2$ concentration and increasing temperatures: https://youtu.be/ONuA9HmkF3M

* Increasing temperatures: https://youtu.be/-V2Uc8Kax_g

* Climate change projections: https://youtu.be/2YE9uHBE5OI

* Climate change: https://www.nelsonguda.com/project/threshold/

* Sea ice loss: https://youtu.be/eYXxAE5grRQ

* Climate data: https://www.jamieperera.com/climate-data-sonification

* Climate data: https://globxblog.github.io/

* Coastal Land Loss : https://datadrivendj.com/tracks/louisiana/

* Other examples in other fields at https://sonification.design/

**Citing these references will be greatly beneficial to my manuscript and I will include them. I have already begun to cite the sonification handbook in my corrections according to your comments below, and I will continue to do so. These other references are highly relevant and I have outlined below that including them will increase the**

**justification for the use of sonification and the manner in which my piece is creatively unique and distinguishable. I plan to use a comparison to increasing temperature sonification as a future potential improvement.**

Line 35: in the sonification handbook (https://sonification.de/handbook/), this is rather termed "indexicality" I believe. In any case I think the author should introduce such concept more thoroughly ("high sonification accuracy to original data" is a bit unclear).

**I have implemented this change and sourced the sonification handbook: Indexicality is a measure of arbitrariness of data mappings, and high indexicality depicts a large degree of conversion accuracy (Hermann, Hunt, & Neuhoff, 2011)**

Line 36: I don't understand what the words "set" and "boundaries" refer to here. Are they elements of the parameter mapping approach used for the right hand? As previously, the author should probably use more space to introduce all these concepts more clearly.

**The word "set" has been replaced by "predetermined" and "and boundaries" has been removed as it does not add to parameters.**

Line 52: explain a bit more what each of these 6 elements represent?

\* linear time: what do you mean exactly?

**Linear time refers to the time of CO2 emissions, and therefore follows the chronological emissions of CO2. This explanation will be put into the introduction and will improve the clarity of communication.**

\* varying length of certain notes: OK but isn't somehow redundant with rhythm, and if not what's the distinction?
**I agree there is little distinction so varying length has been removed, along with instrument as only the piano is used. It has been changed to: "My sonification project uses four elements of sound, namely linear time, frequency, amplitude, and rhythm, to create the *Statistical Composition*."**
\* frequency, amplitude: maybe mention that they are also called pitch and loudness?

**This explanation will be incorporated into this section.**

Line 53: this might be a bit of a controversial statement, but for sure you could state that it carries the formation in an original and engaging way, different from a graph.

**This is an instance of hyperbole that is definitely unnecessary. I have completed this correction in accordance to comments from RC1. I have included "In effect, information is conveyed in an original and engaging way."**

Line 63: What does the "musical backbone" refer to? Key and time signature? please make it explicit. Moreover, it might also be worth explaining that while the score is written in C major, the parameter mapping is not restricted to the 7 notes of the C major scale, but uses all 12 semitones.

**I agree with this improvement. This has been changed to: "This musical backbone incorporates the entire composition being set to the time signature of $\frac{4}{4}$ (a quarter note having the value of a quarter of a measure), the key of middle C, and utilizing all 12 semitones."**

Line 65: The expression "range differentiation" sounds unusual: is it widely used in sonification or in other fields (if so please provide a reference)? "Discretization" sounds more familiar to my ears (https://en.wikipedia.org/wiki/Discretization_of_continuous_features). In any case, the sentence explaining how it's done needs rewording: I guess you meant the difference between the highest and the lowest values?

**This change has been included into the manuscript. I have changed range differentiation into parameter mapping and will include a sufficient definition sourcing the sonification handbook.**

Table S1: the meaning of "Increase of 1" in column "Number of Half Notes" is unclear - why not just give the total number of half-notes used in the parameter mapping, as for other rows? The header "Calculated Interval" is also unclear - maybe "data range covered by one note", if I understood correctly?

**These changes are valuable and will improve the supplement S1. I agree that it is clearer to state the number of half notes used in parameter mapping, and the data range covered by one note. You have understood perfectly and I will implement these changes.**

Lines 72-115: I have to say that I found this description quite confusing and I struggled to understand how the parameter mapping was done exactly, and to distinguish between the "objective" data-driven choices and the "subjective" composition choices. A few suggestions: **Thank you once again for this helpful comment. I am implementing your suggestions as follows below.**

* Maybe the author could adopt a more systematic 2-step structure to describe each movement. Step 1 would be the data-driven parameter mapping: this description should be clear enough to enable a reader to reproduce this part of the score. Step 2 would the composition choices, with the author explaining the intent.

**Such a systematic approach would be a definite improvement, and I thank you for this comment. This will be implemented into the description of each movement. I will describe each composition choice in a full and concise manner, stating intent and effect of each major artistic element. An increase in detail within both the supplement S1 and the text itself is necessary to describe each conversion step. The fact that the general methodology of parameter mapping can be, and is, conducted in slightly different ways makes this challenging, however I will take a basis in the overall structure of the improved supplement S1.**

* I think that both the score and the audio file play the movements described here in the following order: 3-1-2-4-5. Is it correct? If so the movements need to be renamed in the correct order because this is quite confusing in the current state. In any case the author should give the bar numbers associated with each movement.

**The movements are played within the audio file in the order of 1-2-3-4-5, so the order in which they are named is not at fault. I agree that they are not sufficiently separated and distinct, and that both time stamps and bar numbers should be included in the movement titles. I am in the process of changing this in accordance with RC1.**

* If bars 1-8 indeed correspond to movement 3, then I would expect finding 5*12=60 notes (5 years of monthly values). How can these be split into 8 measures? (60/8=7.5). I'm missing something here.

**This is incorrect as bars 1-8 are not in fact movement 3, but movement 1. I have however not explained that the first note is held twice as long to mark the beginning of the song therefore explaining the half beat missing from your calculation. This will be noted in the description of the methodology.**

* I don't understand how the rhythm of bars 47-58 was obtained: is it data-driven or has it been chosen by the author? please clarify.

**For the first 20 years of the movement, each yearly projected average is the length of a quarter note, a fourth of a measure. The 4 years after that are whole notes, and the final year of 2044 is a combined 2 whole notes. This increase in note length is to give a dramatic effect to the ending of the piece. Such an explanation will be included in the description of the methodology of the fifth movement.**

* Is the loudness of the notes somehow data-driven, or is it the author's choice? It's worth explaining for each movement since there are strong dynamics for some of them but not for others - and in terms of perception, dynamics is quite powerful!

**I agree. Dynamics is an artistic choice that is used to create dramaticism. Each movement generally follows the outline of starting soft and becoming increasingly loud. Such dynamics are on the score as well. This will be described for each movement.**

Line 110: I expect scientists working on future greenhouse gases emission and concentrations might not be keen on this best-fit line projection. Why not using the $CO_2$ concentrations for one of the IPCC scenarios? The data are available at this page (probably the same page where the author get the Mauna Loa data): https://www.ipcc-data.org/observ/ddc_co2.html

**This is an optimistic projection, where I am projecting rising $CO_2$ with the assumption that the rate of increase itself will not increase. This is meant to show how even if a positive case is assumed, it is an urgent topic and the music conveys a sense of gloom around this. I will incorporate an evaluation of this method in comparison to the sight**

**you have outlined to verify this, and this can also be mentioned in the suggested improvements section that will be added. This comparison will follow an argumentation that is outlined as follows. My prediction is closest to scenario SSP1-2.6, an optimistic projection of low future emissions. This is visualized in the figure below, however due to copyright this figure will not be included into the manuscript. I will however source this projection and compare 2044 values with my projected CO2 levels and the scenario's. I will source this database here as it is more recently updated:**

**https://www.ipcc.ch/report/ar6/wg1/about/how-to-cite-this-report/.**

[Figure]

NOAA Climate.gov, adapted from IPCC AR6 Technical Summary, Figure TS.4

Line 128-130: this sounds a bit awkward and controversial: the data themselves are numbers, they are not "in English", and some may argue that the language of mathematics is also quite universal, maybe even more than music! In addition, the explanations surrounding these data are indeed in English, but so are the explanations given by the author in this paper. I'm personally not convinced that the point of such data-based musical piece should be to make data more understandable by anyone in the world. In my eyes, its key strength is rather to add an emotional content that is quite unique to music.

**I agree with this comment. Additionally, the first referee commented something similar saying that these two sentences are not necessary and have prominent hyperbole. Therefore, I have deleted these two sentences also in order to highlight the emotional factor of my music in its urgency and gloom that it conveys.**

Line 133: many sonification experiments are physically playable: for just two examples among others, see https://youtu.be/-V2Uc8Kax_g and https://youtu.be/eYXxAE5grRQ

**This is a valid point. Physically playable sonification is not the most universally used forms of sonification, and the creative element makes it unique as each creative choice is made by myself personally. I will distinguish this fact from a unique methodology in general. I will acknowledge the fact that I am not the first to produce a physically playable piece of sonification, and I will acknowledge these two works in the manuscript through references of previous works.**

On the score in table S1:

1. Explicitly identifying the movements would be very useful
**This is understood and agreed that the clarity provided is extremely beneficial. This improvement will be completed.**
2. Wouldn't it be more logical to adapt the time signature and possibly the tempo to the data? 4/4 sounds as logical as any other choice for yearly data, but for decadal averages why not using e.g. 10/4? and e.g. 12/4 for monthly data? (or 12/8 or 12/16? see previous comment on the 5*12 notes of movement 3).
**The movement 3 instance has been addressed previously in this rebuttal, and the fact that the movements are in fact in their chronological order. The fact that this misinterpretation arises however shows the necessary structure and clarity that needs to be improved within my manuscript and product. I found that it was just as easy to change the length of the notes than changing the time signature itself. As this piece is designed to be playable, it is best if it follows somewhat standard formats, and 4/4 is the most standard time signature used in my experience. I have rarely encountered a change of time signature within a song or piece before, and therefore believe such a change would confuse the average musician and increase the level of difficulty significantly. Changing the lengths of the notes around this seemed like a better option, and defining the length value to time period per movement provides a clarity. I believe that it is beneficial to include an explanation to my artistic choices in the supplemental. This will provide room around word count and manuscript length, and provide an added clarity. I will complete this improvement.**

3. A sustain pedal is mentioned for movement 5, it could be added to the score?

**This can and will be added to the score.**

Lines 122-134: I think the author could include a more thorough discussion on both the creation of this piece and any future related work. A few possible discussion topics:

* feedback on the process of turning a fully data-based sonification into a musical arrangement? What were the main challenges faced by the author?

* in many musically-oriented sonification experiments, pitch mapping is restricted to the notes of a particular scale (e.g. C major, E minor etc.). Here the author uses all 12 semitones, which makes it sounds very "chromatic" and quite intriguing. Any feedback on this "chromatic" approach? (motivation, difficulties, etc.)

* any plan to play it live on the piano, since this is mentioned as a original aspect?

* potential interest of complementing the audio with some data visualization/animation or some visual art approach to make such data even more understandable by anyone, irrespective of language?

* Any plan to communicate further on this piece, beyond this paper? (e.g. website, video, blog, art/science/communication events, etc.)

* interest of using more musical instruments?

* etc.

**I agree with this assessment of my work and these topics are all worth mentioning. I additionally will include possibilities of using a similar method with other causes of climate change within the greenhouse gases. There is substantial data of these emissions recorded from the same Mauna Loa Observatory that would be highly relevant. I think the most important discussions that you bring up and mention are the challenges I faced, the use of all 12 semitones, and possibilities of furthering this project through data visualization and animation. I will gladly go into depth on these topics, along with the other highly relevant topics as well, providing the word count allows this.**

Technical comments

Although I'm not a native English speaker, many sentences sound a bit awkward to me: the paper may benefit from an editorial review on this aspect.

**Such a review would be beneficial in improving the communication of this paper. I will conduct such a review myself, and I will have a native English speaker with scientific writing experience review my edited manuscript before submitting the final version. I would of course welcome a similar editorial review on this aspect.**

**I have found this review to be helpful to the highest degree, and I am fully motivated to implement the dominating majority of the suggested improvements. I thank the referee for their diligent work.**